# Optical vectorial-mode parity Hall effect: a case study with cylindrical vector beams

Changyu Zhou [1,5], Weili Liang[1,5], Zhenwei Xie [1,5] ✉, Jia Ma[1], Hui Yang[2], Xing Yang[2], Yueqiang Hu [2,3], Huigao Duan [2,3] ✉ & Xiaocong Yuan [1,4] ✉

The vectorial optical field (VOF) assumes a pivotal role in light-matter interactions. Beyond its inherent polarization topology, the VOF also encompasses an intrinsic degree of freedom associated with parity (even or odd), corresponding to a pair of degenerate orthogonal modes. However, previous research has not delved into the simultaneous manipulation of both even and odd parities. In this study, we introduce and validate the previously unexplored parity Hall effect for vectorial modes using a metasurface design. Our focus lies on a cylindrical vector beam (CVB) as a representative case. Through the tailored metasurface, we effectively separate two degenerate CVBs with distinct parities in divergent directions, akin to the observed spin states split in the spin Hall effect. Additionally, we provide experimental evidence showcasing the capabilities of this effect in multi-order CVB demultiplexing and parity-demultiplexed CVB-encoded holography. This effect unveils promising opportunities for various applications, including optical communication and imaging.

The optical field plays a crucial role in the design of photonic devices, offering multiple degrees of freedom for manipulation[1]. One significant aspect of the optical field is its polarization, which has extensively been exploited in various disciplines within photonics. In contrast to the scalar optical field, which has uniformly distributed states of polarization (SOPs), vectorial optical fields (VOFs) enhance light-matter interactions due to their spatially variant SOPs[2,3]. Such VOFs, originated from natural solutions of the vectorial Helmholtz equation[4], can also be generated by the combination of Laguerre-Gaussian (LG), Hermite-Gaussian, or Bessel-Gaussian mode[2]. Common examples include vector beams such as cylindrical vector beam (CVB)[5], Poincaré beam[6,7], and vector HE mode in an optical fiber. VOFs have gained notable research attentions in recent years and found a range of applications, e.g., utilization of CVBs in optical microscopy[8], optical communication[9,10] and optical trapping[11,12]. These exhibit the powerful manipulating means and promising foregrounds by the utilization of VOFs.

Recently developed optical metasurfaces has enriched the research on the manipulation of optical fields, with the advantage of exhibiting point-by-point polarization response across the transverse plane at subwavelength scale[13–15]. By adjusting dimensions of the meta-atoms, vectorial manipulation is achieved through locally effective birefringence[16,17], thereby modifying the local SOP of the impinging waveform. This form-birefringence-based metasurface has found wide applications in meta-lenses[18–21], structured light field manipulation[22–24], and holography[25–28], providing a promising platform for manipulating VOFs at the nanoscale. One notable characteristic of VOFs is their presence of a pair of degenerate modes with different polarization mode parities. For instance, the 1st-order radially polarized CVB possesses even parity, while its azimuthally polarized counterpart has odd parity. The existence of the parity in VOFs, in addition to the polarization topology, presents potential applications in photonics. However, research on manipulating the parity of VOFs remains largely unexplored.

[1]Nanophotonics Research Center, Institute of Microscale Optoelectronics & State Key Laboratory of Radio Frequency Heterogeneous Integration, Shenzhen University, Shenzhen, China. [2]National Research Center for High-Efficiency Grinding, College of Mechanical and Vehicle Engineering, Hunan University, Changsha, China. [3]Greater Bay Area Institute for Innovation, Hunan University, Guangzhou, Guangdong Province, China. [4]Research Centre for Frontier Fundamental Studies, Zhejiang Lab, Hangzhou, China. [5]These authors contributed equally: Changyu Zhou, Weili Liang, Zhenwei Xie. ✉e-mail: ayst31415926@szu.edu.cn; duanhg@hnu.edu.cn; xcyuan@szu.edu.cn

In this study, we analyze and demonstrate that two degenerate vectorial modes with different parities can be arbitrarily modulated using a tailored form-birefringence metasurface. Without loss of generality, we adopt widely used CVBs for the proof-of-concept demonstration in this study. We experimentally demonstrate the separation of the mode parities for ±1st-order and 2nd-order CVBs, thereby realizing the parity Hall effect, analogous to the spin-states split observed in the well-known photonic spin Hall effect[29–33]. However, unlike the spin-Hall effect, which is limited to two spin states, the parity-Hall effect applies universally to any other VOFs (e.g., CVBs and Poincaré beams) with parity. This offers a potent avenue to realize a range of multifunctional applications in photonics. For the demonstration of its powerful capacity, we also experimentally achieve both the topological charge and the parity demultiplexing up to $7 \times 2 = 14$ channels using a single form-birefringence metasurface. Furthermore, we experimentally demonstrate the application of parity-demultiplexed CVB-encoded holography for ±2nd-order and ±3rd-order CVBs. We anticipate that this optical vectorial-mode parity Hall effect will inspire research in multi-dimension manipulation of the optical field, enabling the design of multi-functional metadevices, and further stimulating a range of applications such as in optical communication and imaging.

## Results

### Concept and design

To explain the concept of parity for Vector Optical Fields (VOFs), we begin with a general expression derived from the superposition of states on the hybrid-order Poincaré sphere, as presented in[2]

$$\Psi_{LG_p^{m_1,m_2}}(\mathbf{r}) = \cos\left(\frac{\phi}{2}\right) LG_p^{m_1} e^{i\theta/2} \hat{\mathbf{e}}_R + \sin\left(\frac{\phi}{2}\right) LG_p^{m_2} e^{-i\theta/2} \hat{\mathbf{e}}_L, \quad (1)$$

in the LG basis, where $\phi$ and $\theta$ are the orientation angle and ellipticity angle on the sphere, respectively, $m_1$ ($m_2$) represents the azimuthal index

(topological charge), and $p$ is the radial index of the LG modes. $\hat{\mathbf{e}}_R$ and $\hat{\mathbf{e}}_L$ represent the right and left circular polarizations, respectively. Without loss of generality, we choose LG basis for the purpose of our demonstration. The parity of a VOF is determined by the polarization distributions in the transverse plane of the beam and can be defined using the parity operator $\hat{P}\boldsymbol{\psi}(\hat{P}\mathbf{r}) = P\boldsymbol{\psi}(\mathbf{r})$, where $\boldsymbol{\psi}(\mathbf{r})$ is the SOP of a VOF, $P = \pm 1$ is the corresponding eigenvalue. The VOFs on the Poincaré sphere can be simply expressed as a superposition of $\boldsymbol{\Psi}(\mathbf{r}) = \exp(im_+\varphi)\boldsymbol{\psi} = \exp(im_+\varphi)[\exp(im_-\varphi) \quad \exp(-im_-\varphi)]^T$ and $\exp(im_+\varphi)[i\exp(im_-\varphi) \quad -i\exp(-im_-\varphi)]^T$ in the $\hat{\mathbf{e}}_R$-$\hat{\mathbf{e}}_L$ basis, where $\varphi$ represents the azimuthal angle in the transverse plane of the beam, and $m_\pm = (m_1 \pm m_2)/2$. By determining the eigenvalue $P = 1$ or $-1$ for the above SOPs (details can be found in the supplementary materials S1), we can establish the parity. For instance, when $m_1 = -m_2 = 1$, the radially polarized CVB has even parity ($P = 1$), while the azimuthally polarized CVB has odd parity ($P = -1$).

The design mechanism of the vectorial-mode parity Hall effect is based on the local form-birefringence effect[17]. In a conventional single-axis birefringent crystal, two orthogonal directions exhibit different principal refractive indices, corresponding to two orthogonally polarized states. Consequently, when the normally incident polarized light enters the crystal, its velocity depends on the incident SOP (Fig. 1a). Alternatively, it is possible to create an effective form birefringence using a homogeneous array of ordered unit cells in the metasurface with varying feature sizes along the two orthogonal axes to emulate the same polarization dispersion (Fig. 1b). However, conventional polarization-dependent birefringent materials only respond to homogenous polarization and are therefore unsuitable for VOFs with spatially variant SOPs. Based on the aforementioned analysis, it becomes apparent that the parity (even or odd mode) related birefringence can be easily attained through a symmetrically distributed birefringent-metasurface configuration, as depicted in Fig. 1c. Ultimately, the parity Hall effect can be achieved by tailoring the spatially varying version of the symmetrically distributed form birefringence (Fig. 1d).

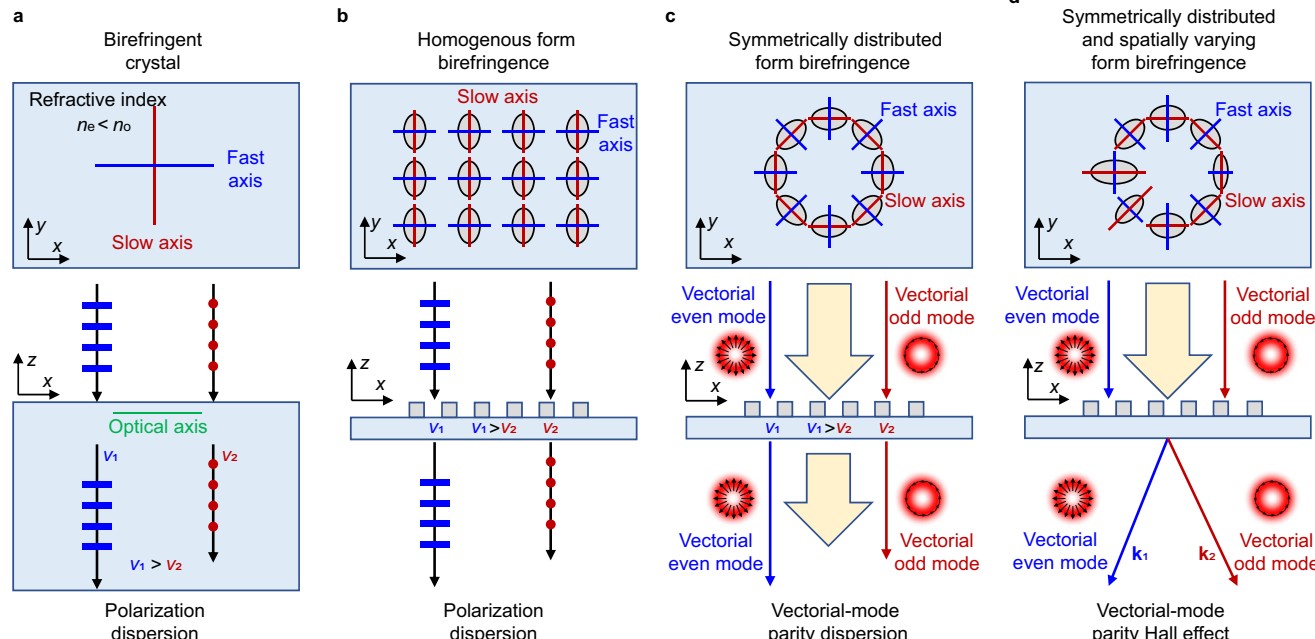

**Fig. 1 | Underlying physical mechanisms of the vectorial-mode parity Hall effect. a** Schematic representation of polarization dispersion in a conventional birefringent crystal. $n_o$ ($n_e$) and $v_1$ ($v_2$) represent the effective refractive index and propagating velocity for the ordinary (extraordinary) wave, respectively. **b** Achieving polarization dispersion with homogenous form birefringence. **c** Enabling vectorial mode parity dispersion through a symmetric form-birefringence configuration. **d** Realization of the vectorial-mode parity Hall effect by introducing spatial variation to the configuration in **c**. $\mathbf{k}_1$ and $\mathbf{k}_2$ represent the wave vectors of the output two vectorial modes, respectively.

Metasurfaces inherently offer a platform to achieve such an effect by virtue of their capacity to manipulate the local SOPs in VOFs. The designed metasurface comprisesa periodic array of the dielectric elliptical resonators made of Titanium Dioxide (TiO$_2$), positioned on a silica substrate. The elliptical shape induces birefringence in the resonators, which can be locally characterized using the Cartesian coordinate form of the Jones matrix[16,17]

$$\tilde{T} = \tilde{R}[-\varphi(x,y)]\begin{pmatrix} e^{i\Omega_+(x,y)} & \\ & e^{i\Omega_-(x,y)} \end{pmatrix}\tilde{R}[\varphi(x,y)], \quad (2)$$

where $\tilde{R}(\varphi)$ is the 2 × 2 rotation matrix and $\Omega_\pm$ represents the modulation phase imparted by the resonators. In the actual metasurface design, $\varphi(x,y)$ in the Eq. (2) is customized to match the distribution of optical field polarizations from the input VOFs.

A crucial step in achieving the parity Hall effect is the application of the generalized Snell's law of refraction[13]

$$\sin(\theta_t)n_t - \sin(\theta_i)n_i = \frac{\lambda_0}{2\pi}\cdot\frac{d\Omega_P}{dq} = P\frac{\lambda_0}{2\pi}\cdot\frac{d\Omega}{dq}, \quad (3)$$

where $\theta_i$ and $\theta_t$ are the input and output angles of the beam, respectively, $n_i$ and $n_t$ represent the refractive indices of the two media, $\lambda_0$ is the vacuum wavelength, and the symbol '$q$' denotes an arbitrary direction in the $(x, y)$ plane. Compared to the commonly used scheme in anomalous refraction[13,34,35]. The difference in Eq. (3) is the introduced parity-dependent deflection momentum imparted by the resonators, represented by the modulation phase $\Omega_P$, which determines the separation of even and odd modes of a VOF to different directions. In this work, we restrict our demonstrations with the CVBs, e.g., on the 1st-order Poincaré sphere, there exist only two modes with parity, one is the radially polarized CVB with even parity ($P = 1$), and the other one is the azimuthally polarized CVB with odd parity ($P = -1$). Using the form-birefringence metasurface design above, the even and odd modes of a VOF can be separated to different directions, as illustrated in Fig. 2. This method can be also generalized to the other optical vectorial modes, e.g., Poincaré beams.

## Verification of the vectorial-mode parity Hall effect

The basic meta-atom of the metasurface consists of a TiO$_2$ elliptical nanopillar, which acts as an effective birefringent resonator that modulates the phase and magnitude of incident light. The period of a meta-atom is 500 nm, and the height of the elliptical nanopillar is 1000 nm. In order to achieve the desired distributed phase modulation $\Omega_{\pm x}$ at the working wavelength of 633 nm (Fig. 3a), the lengths of the two axes of the elliptical nanopillars are varied (see supplementary materials S2 for details). The incident CVB of vectorial mode $|m\rangle$ is then modulated by the parity-dependent gradient phase and output with a designed deflection (refractive) angle, as explained in Eq. (3).

The actual device was fabricated on a silica plate, and the scanning electron microscope (SEM) images are shown in Fig. 3b. For the experimental setup, a commercial 632.8-nm He-Ne laser was selected to

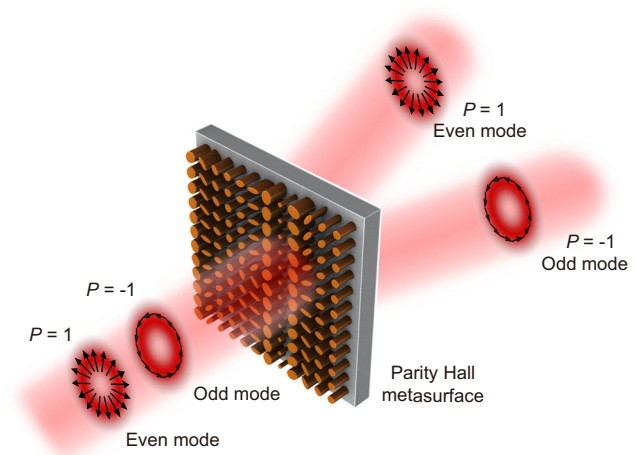

**Fig. 2 | Metasurface enabled vectorial-mode parity Hall effect.** The even ($P = 1$) and odd ($P = -1$) modes of a CVB are separated into different directions by a form-birefringence metasurface.

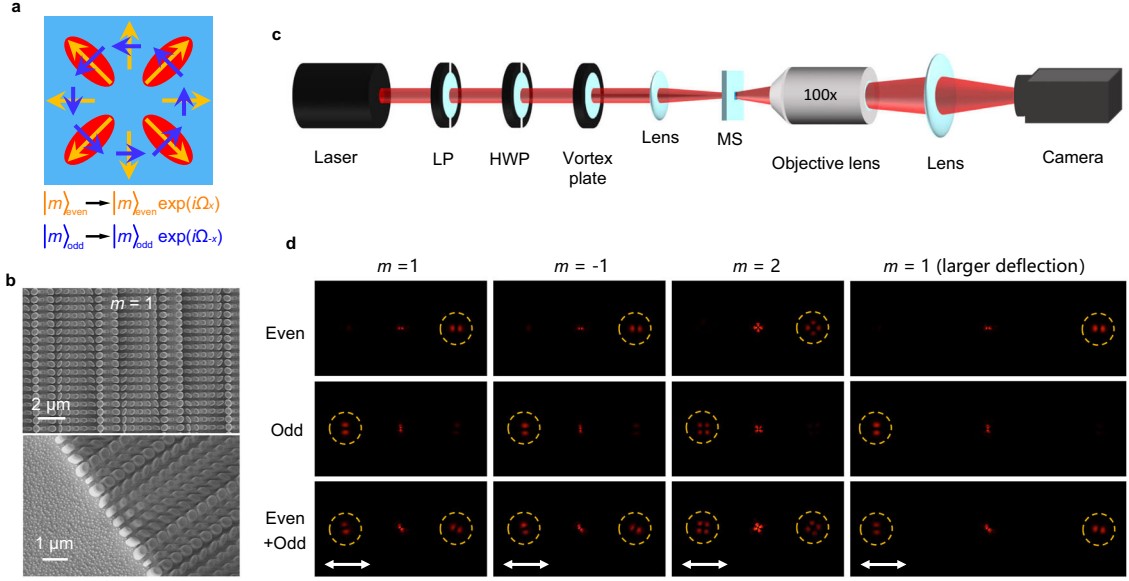

**Fig. 3 | Experimental verification for the parity Hall effect. a** Schematic of the desired phase modulation $\Omega_{\pm x}$ for even $|m\rangle_{even}$ and odd $|m\rangle_{odd}$ CVB modes. **b** SEM images of the fabricated sample. **c** Experimental setup. LP linear polarizer, HWP half wave plate, MS metasurface. **d** Experimentally tested results for demonstrating the separation of the even and odd CVBs with topological charges $m = 1$, $-1$, and 2. The presence of the white double arrows at the bottom indicate that the resulting field distributions are obtained by applying a horizontal polarizer.

test the designed metasurface. The desired CVB was generated by the vortex plate thereafter, while a half wave plate (HWP) placed in front of the vortex plate was used to control the vectorial mode (even, odd, or both) of the generated CVB. The output light was recorded by a CMOS camera (Fig. 3c). In our design, the deflection angle for an even (or odd) CVB is 9.1°, according to the Eq. (3). The experimental results demonstrate the separation of the ±1st-order radial (even) and azimuthal (odd) polarized CVB modes along the horizontal $x$ direction (Fig. 3d). The same results are obtained for higher-order CVBs, as we have demonstrated with the 2nd-order CVB in our experiment. The actual measured deflection angles are 9.02°, 9.02°, and 8.92° for $m = 1$, −1, and $m = 2$, respectively. In addition, the parity Hall effect is also evident for a larger deflection angle (Fig. 3d), where we obtained the measured deflection angle of 18.33°, which is very close to the designed value of 18.45°.

## Demultiplexing of the parity and topological charge in multiple vector modes

CVBs possess significant multiplexing and demultiplexing potential due to their unlimited topological charges[36–38]. Building upon the vectorial-mode parity Hall effect, we introduce an efficient scheme that enables the separation of both the parities and topological charges of the CVBs, thereby significantly expanding the potential for multi-channel and multi-function applications.

In addition to the polarization distributions of the vectorial modes, we observe that the spatial intensity distributions for two CVBs with different topological charges are distinct. Specifically, the intensity distributions exhibit a donut-like pattern that deviates from the geometric origin as the topological charge of the CVB increases (Fig. 4a). Motivated by this phenomenon, we utilize the power-normalized intensity distributions of the $m$th-order CVB (depicted in Fig. 4a) as the references for arranging the responsive resonators in the entire metasurface plane. These arrangements of resonators are determined by the probability $p_m(r) = \sum_{i=0}^{6} I_m(r)/I_i(r)$, where the probability $p_m(r)$ is selected to represent the likelihood of the $m$th-order resonators at a given distance $r$ from the center of the metasurface. Additionally, $I_m(r)$ denotes the light intensity of the $m$th-order CVB at the same distance $r$. The arrangement of elliptical resonators on the metasurface is visualized in Fig. 4b, highlighting their approximate alignment with the light intensity distributions of the corresponding-order CVB depicted in Fig. 4a.

Combined with the parity Hall effect, each CVB mode is designed to deflect to different directions with different imparted gradient phases, as described in Eq. (3). To recognize and distinguish the CVBs in the experiment, we further introduce the lens phase and the compensatory vortex phase with an inverse topological charge to the metasurface (see supplementary materials S3 for details). This conversion allows us to focus the desired demultiplexed $m$th-order CVB into a Gaussian spot. In our design, the topological charge- and parity-demultiplexed CVBs are evenly distributed on a circle (Fig. 4c). Our experimental results (Fig. 4d) demonstrate successful vectorial mode demultiplexing up to 14 channels, with the demultiplexed topological charges ranging from $m = 0$ to $m = 6$, and also the even and odd parities.

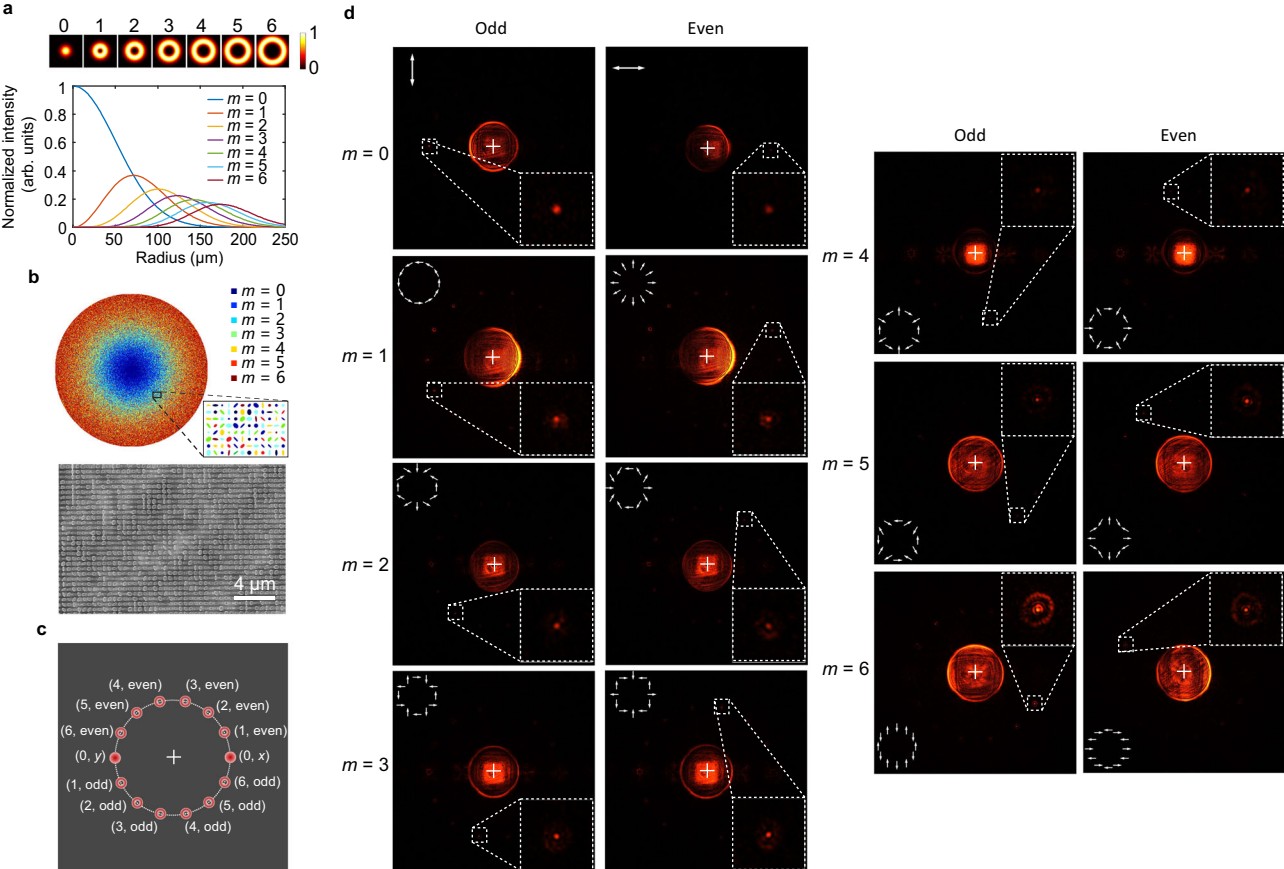

**Fig. 4 | Parity and topological charge demultiplexing of multiple CVBs. a** The spatial intensity distributions of CVBs with topological charge from $m = 0$ to $m = 6$. **b** The arrangements of elliptical resonators on the metasurface and the SEM image of the fabricated sample. **c** Schematic of distributed demultiplexed results of CVBs with both topological charge and parity, denoted by ($m$, parity). **d** Experimentally tested results of CVB demultiplex up to 14 channels in the experiment with the demultiplexed two parities and topological charges from $m = 0$ to $m = 6$. The white-arrows pattern in each subgraph indicates the polarization distributions of the corresponding CVB.

## Parity and topological charge demultiplexed CVB meta-hologram

Metasurfaces can also be utilized for high-volume holographic imaging. Recent studies have demonstrated the utilization of orbital angular momentum (OAM) of light to achieve the OAM-preserved holography[39–41]. By leveraging the encoded OAM selectivity on the metasurface, an encoded image can only respond to the corresponding topological-charge OAM beam, enabling OAM demultiplexed holography. In this study, we extend this method further by encoding both the topological charge and the parity of the CVB into the hologram, achieving the parity and topological charge demultiplexed holography. Figure 5a illustrates the schematic design of such an approach. The target image is sampled using a two-dimensional dot matrix, with a sampling distance of $d = 3\,\mu m$ to avoid interference between the reconstructed CVBs. Subsequently, a CVB-preserved hologram is obtained through inverse Fourier transform. Similar to the OAM-selected holography, the CVB mode-selected holography can be achieved by introducing an inverse-topological-charge vortex phase into the hologram (Fig. 5b). As a proof of concept, we experimentally showcase an eight-channel CVB demultiplexed holography with the demultiplexed topological charges of $m = \pm 2, \pm 3$, as well as the even and odd parities. In our design, four nanopillars constitute a

metamolecule, each of them responds to different CVB modes (Fig. 5b). The microscopy and SEM images of the fabricated sample are presented in Fig. 5c. The experimental results for CVB demultiplexed metasurface holography are depicted in Fig. 5d, where the desired CVBs reconstruct holographic digits "1, 2, 3, 4" for $m = \pm 2$ channels, and letters "A, B, C, D" for $m = \pm 3$ channels at different positions. The results demonstrate the designed metasurface can effectively demultiplex CVB holograms with different orthogonal modes.

## Discussion

In summary, we propose and demonstrate the optical vectorial-mode parity Hall effect using a form-birefringence metasurface. By manipulating the local SOPs through the resonators in the metasurface, the induced modulation phase produces additional deflection momentums whose directions depend on the vectorial mode's parity, achieving a parity-dependent split of VOFs. As evidence of the proposed method, we successfully separated even and odd parities for ±1st- and 2nd-order CVBs in experiments. Furthermore, we demonstrated that this effect is capable of achieving multiple functions, including the topological charge and parity demultiplexing up to 14 channels and the CVB-encoded holography for ±2nd- and ±3rd-order CVBs. Additionally, this optical vectorial-mode parity Hall effect can be

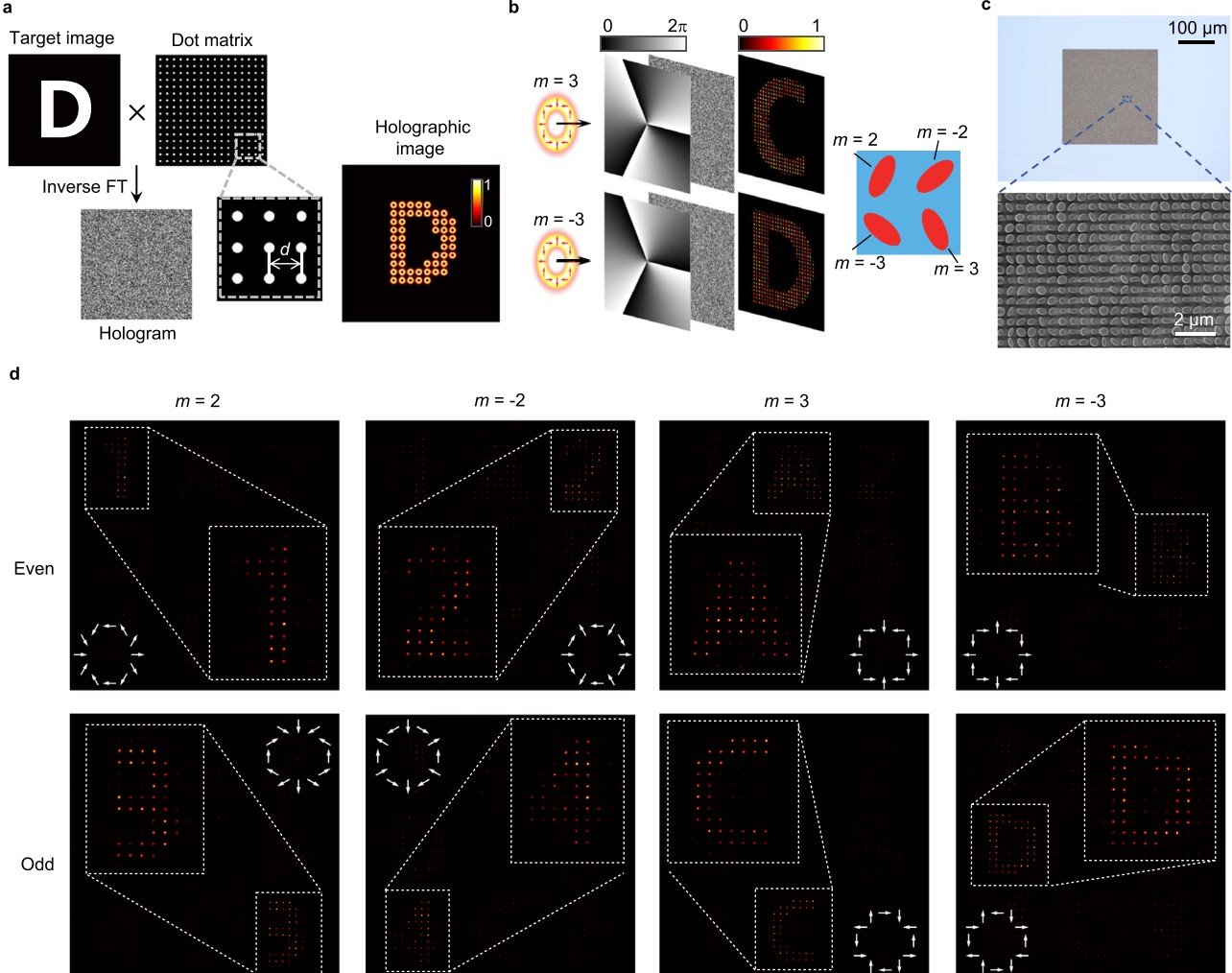

**Fig. 5 | CVB-encoded parity-demultiplexed holography. a** Schematic of the procedure for generating a desired CVB-encoded hologram, together with generated the holographic image. The sampling distance of the dot matrix is $d = 3\,\mu m$. FT: Fourier transform. **b** Schematic of the CVB-selective holography using a metasurface encoded hologram, along with the design of its basic metamolecule with

demultiplexed topological charges of $m = \pm 2, \pm 3$. **c** Microscopy and SEM images of the fabricated sample. **d** Experimentally tested results of the CVB demultiplexed holography with topological charges of $m = \pm 2, \pm 3$, and also the even and odd parities. The white-arrows pattern in each subgraph indicates the polarization distributions of the corresponding CVB.

extended to the optical fiber platform with a form-birefringence device integrated on the facet of a cleaved optical fiber[42], beyond free-space applications. Such an effect can also be applied to non-linear domains. However, it is noteworthy that the original modulation of the meta-surface would be affected by the non-linear response of materials, causing a perturbation to the designed performances. Hence the chosen parameter for the metasurface should be carefully tailored.

The proposed optical vectorial-mode parity-Hall metasurfaces expand the degrees of freedom in light field steering and have immense potential for a wide range of applications. By designing and fabricating metasurfaces with precise control over their structural parameters, it becomes possible to achieve accurate control over the parity of VOFs. Furthermore, the proposed parity Hall effect in this work is universal and can be extended to other frequency ranges, such as the terahertz regime and radio frequency. Therefore, the corresponding design should be adaptable to the desired frequency for achieving the desired modulation. This fine-grained manipulation of optical fields could lead to numerous exciting applications, such as breakthroughs in super-resolution imaging, optical information processing, optical sensing, and high-speed optical communication. Overall, the research on optical vectorial-mode parity-Hall meta-surfaces is highly promising and has the potential to revolutionize many aspects of modern optics and photonics.

## Methods
### Sample fabrication
The samples were fabricated using the electron beam lithography (EBL) in combination with the etching technique. Initially, a 1000-nm-thick polymethyl methacrylate (PMMA) electron-beam resist layer was spin-coated onto a transparent silica substrate with an ITO film layer. This coated sample was then baked on a hot plate at 180 °C for 4 min. Subsequently, the sample was exposed by the EBL using a voltage of 100 kV and a beam current of 200 pA. After exposure, the sample was immersed in a mixed solution of isopropanol and methyl isobutyl ketone (IPA: MIBK = 3:1) for 1 min, followed by fixation in the IPA solution for an additional minute at room temperature. The exposed area of the sample was then filled with a 220 nm $TiO_2$ layer using the atomic layer deposition (ALD) system. For the positive photoresist exposure process, we utilized PMMA, which was voided before deposition. The deposited thickness of $TiO_2$ was determined by the semi-minor axis of the maximum meta-atom. Following this step, a 1000-nm thick layer of $TiO_2$ covered the entire sample, which was subsequently removed using ion beam etching (IBE) in the subsequent process. Once the top layer of $TiO_2$ had been removed, reactive ion etching (RIE) was employed to eliminate the resistance. As a result, $TiO_2$ nanostructures with a high aspect ratio (up to 10) were obtained.

### Numerical simulations
Numerical simulations of the metasurfaces were performed using the commercial software Lumerical FDTD Solutions, which is based on the finite difference time domain method. The period of meta-atoms was set to 500 nm. For the simulations, perfectly matched layers (PML) were utilized to accurately calculate the behavior of the meta-holograms. The substrate was incorporated into the simulations, and the refractive index of $SiO_2$ was assumed to be 1.46 at the operating wavelength of 632.8 nm. The refractive index of $TiO_2$ was determined through measurements using an ellipsometer.

## Data availability
The data supporting the findings of this study are available in the Supplementary Information

## Code availability
All numerical codes are available upon request from the corresponding authors.

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

## Acknowledgements

This research was supported by the Guangdong Major Project of Basic and Applied Basic Research No. 2020B0301030009; National Natural Science Foundation of China (61935013, 62375181, 61975133); Shenzhen Science and Technology Program (JCYJ20200109114018750); Shenzhen Peacock Plan (KQTD2017033011044030); Scientific Instrument Developing Project of ShenZhen University (No.2023YQ001); Shenzhen University 2035 Initiative (No.2023B004). Huigao Duan thanks the financial support from the National Natural Science Foundation of China (52221001) and Shenzhen Science and Technology Program (JCYJ20220530160405013). The authors would like to acknowledge the Photonics Center of Shenzhen University and Professor Shumin Xiao from Harbin Institute of Technology (Shenzhen) for technical support in device fabrication. Changyu Zhou thanks Peng Wu for the helpful discussions.

## Author contributions

C.Z. conceived the original idea. Z.X. proposed the concept. W.L., C.Z., and Z.X. carried out the calculations and simulations. W.L. conducted the measurements. J.M. participated in the partial measurements. H.Y., X.Yang, Y.H., and H.D. fabricated the samples. C.Z. wrote the manuscript. W.L., Z.X., H.D., and X.Yuan revised the manuscript. Z.X., H.D., and X.Yuan supervised the project. All the authors discussed the results and commented on the manuscript.

## Competing interests

The authors declare no competing interests.
