## [Peer Review File · Nature Communications]

REVIEWER COMMENTS

Reviewer #1 (Remarks to the Author):

A birefringence metasurface design is used to record Hall effect for the vectorial optical field (VOF). Two cylindrical vector beams (CVB) with different parities are separated into different directions, similarly to what is happening in the Hall effect. The authors claim that their concept can be generalized to multi-order cylindrical vector beams and act as a multiplexing/demultiplexing setup. The underlying physical mechanism as well the experimental layout. The alignment between the spatial signature of the field and the order of cylindrical vector beam is demonstrated by measurements. The procedure for generating a desired CVB-encoded hologram is also described. The paper entertains an interesting idea applied in analog signal processing, imaging, and optomechanics; it experimentally verifies it in the lab. However, to penetrate in a publication venue as Nature Communications requires more effort. In particular:

(A) A stronger novelty statement is required. How this work is different with other recent studies [1-3] with similar purpose?

(B) The authors should explain how the proposed idea will change if nonlinearities were taken into account. How bistability would modify the multiplexing utility?

(C) The connection of this concept with usual anomalous diffraction gratings [4,5] should be properly elaborated. A compare and contrast analysis between the two mechanisms should be provided in the revised version.

(D) The performance of the proposed device in terms of resolution or angular separation, should be connected with the structural parameters of the design (size and density of elliptical nanopillars etc).

[1] Spin Hall Effect of Double-Index Cylindrical Vector Beams in a Tight Focus, Micromachines, 2023.

[2] Actively manipulating asymmetric photonic spin Hall effect with graphene, Carbon, 2020.

[3] Spin Hall effect of transversely spinning light, Science Advances, 2022.

[4] Anomalous refraction into free space with all-dielectric binary metagratings, Physical Review Research, 2020.

[5] Anomalous refraction and reflection characteristics of bend V-shaped antenna metasurfaces, Scientific Reports, 2019.

Reviewer #2 (Remarks to the Author):

The authors present a case study on vectorial-mode parity Hall effect by use of a form-birefringence metasurface resonating at optical frequencies. Experimental manipulation of the parities of two cylindrical vector beams via the designed metasurface has reassembled the well-known spin Hall effect. In addition, the authors have demonstrated parity-demultiplexed cylindrical vector beam-encoded holography. The findings are interesting and significant, and the presentation is well organized. The manuscript can be recommended for publication in Nature Communications, provided the authors consider the following points.

1. The vectorial-mode parity Hall effect demonstrated here can be extended to the optical fiber platform beyond free space applications. Can this effect be extended to other frequency ranges, such as terahertz and microwave regimes? Are there any potential complications?

2. The contrast of the measured holographic images shown in Fig. 5d appears low compared to that in Fig. 5a. It would be good if the authors can add a brief discussions on this.

3. How the aspect ratio of the TiO₂ nanostructures (up to 10 here) affects the demonstrated Hall effect, parity-demultiplexing, and meta-hologram?

** See Nature Portfolio's author and referees' website at www.nature.com/authors for information about policies, services and author benefits.

RESPONSE LETTER

REVIEWER COMMENTS

Reviewer #1 (Remarks to the Author):

A birefringence metasurface design is used to record Hall effect for the vectorial optical field (VOF). Two cylindrical vector beams (CVB) with different parities are separated into different directions, similarly to what is happening in the Hall effect. The authors claim that their concept can be generalized to multi-order cylindrical vector beams and act as a multiplexing/demultiplexing setup. The underlying physical mechanism as well the experimental layout. The alignment between the spatial signature of the field and the order of cylindrical vector beam is demonstrated by measurements. The procedure for generating a desired CVB-encoded hologram is also described.

The paper entertains an interesting idea applied in analog signal processing, imaging, and optomechanics. It experimentally verifies it in the lab. However, to penetrate in a publication venue as Nature Communications requires more effort. In particular:

Reply: We would like to express our sincere appreciation for invaluable feedback on the manuscript. In response to your concerns, we are trying our best knowledge to provide comprehensive and detailed replies in the following sections, respectively.

(A) A stronger novelty statement is required. How this work is different with other recent studies [1-3] with similar purpose?

Reply: We would like to emphasize that the novelty of our work lies in the fact that the parity-Hall effect for vectorial optical fields (VOFs) is firstly found and reported in this manuscript. The parity in VOFs represents a distinct degree of freedom (DOF) from its counterpart in terms of spin, and it can be described on the general hybrid-order Poincaré sphere (Eq. 1 in the manuscript) with a different position. The parity in this case is associated with the symmetry exclusively comparing to the spin. It is noted that the suggested references [1-3] are

background work oriented by the spin Hall effect, nevertheless, we are grateful to the reviewer for useful and comparative references and we have added them to the references list.

Our proposed method involves the use of a metasurface-based approach to separating two parities of VOFs, using circular vector beams as an example in this work. To the best of our knowledge, this effect has not been previously reported in literature. It envisions a new physical freedom to the Hall effect that is distinct from the well documented spin-Hall effect as described in Ref. (1-3) and others. Additionally, the proposed parity-Hall effect is not only restricted to a single VOF (e.g., the 1st-order CVB), but also extended to any other high-order VOFs, making it superior to the spin-Hall effect with only two spin states. Following the suggestions, we have further highlighted the point with stronger elaborations in terms of novelty in the revised manuscript.

Changes made in the revised manuscript:

Page-1

Manipulating the optical field is a primary objective in the design of photonic devices. This is achieved by utilizing various degrees of freedom (DOFs), such as wavelength, phase, polarization, magnitude, and even the vectorial mode of light. The vectorial optical field (VOF) holds particular significance in light-matter interactions due to its spatially varying states of polarization. It is worth noting that, in addition to its inherent polarization topology, the VOF also possesses an inherent DOF of parity (even or odd), which is associated with a pair of degenerate orthogonal modes. However, to the best of our knowledge, no previous studies have explored the manipulation of both the even and odd parities of an optical field. Harnessing the DOF of parity could open up a wide range of applications for the manipulation of optical fields. In this study, we propose and demonstrate the previously unidentified parity Hall effect for vectorial modes using a form-birefringence metasurface design. As a proof of concept, we utilize a cylindrical vector beam (CVB) as a case study. The designed metasurface effectively separates two degenerate orthogonal CVBs with different parities into different directions, analogous to the split of spin states in the well-known spin Hall effect. Furthermore, we experimentally demonstrate the capabilities of this effect in multi-order CVB demultiplexing and parity-demultiplexed CVB-encoded holography. We envision that

this effect holds promising prospects for a wide range of applications, including optical communication, imaging, and optomechanics.

Page-2

... analogous to the spin-states split observed in the well-known photonic spin Hall effect [29-33]. However, unlike the spin-Hall effect, which is limited to two spin states, the parity-Hall effect applies universally to any other VOFs (e.g., CVBs and Poincaré beams) with parity. This offers a potent avenue to realize a range of multifunctional applications in photonics. For the demonstration of its powerful capacity, we also experimentally achieve both the topological charge and the parity demultiplexing up to $7 \times 2 = 14$ channels using a single form-birefringence metasurface.

Page-12

- [31] L. Peng, H. Ren, Y.-C. Liu, T.-W. Lan, K.-W. Xu, D.-X. Ye, *et al.* Spin Hall effect of transversely spinning light. *Science Advances*, **8**, eabo6033 (2022).
- [32] Y. Wu, L. Sheng, L. Xie, S. Li, P. Nie, Y. Chen, *et al.* Actively manipulating asymmetric photonic spin Hall effect with graphene. *Carbon*, **166**, 396-404 (2020).
- [33] A. A. Kovalev, V. V. Kotlyar. Spin Hall Effect of Double-Index Cylindrical Vector Beams in a Tight Focus. *Micromachines*, **14**, 494 (2023).

(B) The authors should explain how the proposed idea will change if nonlinearities were taken into account. How bistability would modify the multiplexing utility?

Reply: We appreciate your insightful comment. In our work, all processes are based on linear effect and they have shown excellent performance in the current scope of research. The prospect of introducing nonlinearity to the device presents an intriguing and challenging further step. Given that the performance of the designed metasurface, including aspects like multiplexing and holography, heavily relies on the phase modulation of the nanopillars, incorporating nonlinear effects such as harmonic generation or bistability could have implications for intensity, frequency, phase modulation, and even the mode's parity, potentially disrupting the intended performances. While a more in-depth analysis is beyond the scope of this paper, we believe that it would be appropriate to give a brief discussion on the nonlinearity

topic as suggested by the reviewer.

Changes made in the revised manuscript:

Pages-8&9

beyond free-space applications. Such an effect can also be applied to nonlinear domain. However, it is noteworthy that the original modulation of the metasurface would be affected by the non-linear response of materials, causing a perturbation to the designed performances. Hence the chosen parameter for the metasurface should be carefully tailored.

(C) The connection of this concept with usual anomalous diffraction gratings [4,5] should be properly elaborated. A compare and contrast analysis between the two mechanisms should be provided in the revised version.

Reply: We sincerely appreciate the suggestion pointed out by the reviewer. The proposed method for separating two parities of VOF is based on the general Snell's (refractive or reflective) law in Eq. (3) of the manuscript, which is similar to that of, e.g., Ref. [5]. Generally, all of these (including Ref. [4-5] and our work) are based on the introduced extra momentum along the deflected direction, eventually causing an anomalous refraction/reflection. However, the difference in our work (compared to Ref. [4-5] or other related works) is the extra parity-dependent term P and the parity-dependent phase gradient in Eq. (3).

(3)

We follow your advice and discuss these mechanisms in the revised manuscript as below.

Changes made in the revised manuscript:

Pages-4&5

Compared to the commonly used scheme in anomalous refraction [13, 34, 35]. The difference in Eq. (3) is the introduced parity-dependent deflection momentum imparted by the resonators, represented by the modulation phase Ω_P , which determines the separation of even and odd modes of a VOFs to

different directions.

Page-12

[34] Y. Xie, C. Yang, Y. Wang, Y. Shen, X. Deng, B. Zhou, *et al.* Anomalous refraction and reflection characteristics of bend V-shaped antenna metasurfaces. *Scientific Reports*, **9**, 6700 (2019).

[35] N. L. Tsitsas, C. Valagiannopoulos. Anomalous refraction into free space with all-dielectric binary metagratings. *Physical Review Research*, **2**, 033526 (2020).

(D) The performance of the proposed device in terms of resolution or angular separation, should be connected with the structural parameters of the design (size and density of elliptical nanopillars etc).

Reply: Thank you for your helpful suggestions. We have included the detailed parameters (size parameters of elliptical nanopillars) in the revision, which can be found in the revised Supplementary Materials.

Changes made in the revised Supplementary Materials:

Tables-S1&S2

Table S1. Parameters of D_x (nm)

$\phi_x \backslash \phi_y$	$-3\pi/4$	$-\pi/2$	$-\pi/4$	0	$\pi/4$	$\pi/2$	$3\pi/4$	π
$-3\pi/4$	400	208	240	280	348	420	280	112
$-\pi/2$	156	188	220	248	300	384	420	100
$-\pi/4$	148	180	204	232	268	332	420	100
0	140	172	196	220	256	308	408	420
$\pi/4$	132	160	188	208	240	288	372	420
$\pi/2$	128	148	176	196	224	264	340	420
$3\pi/4$	420	148	164	180	204	240	304	372
π	212	272	276	180	200	228	284	136

Table S2. Parameters of D_y (nm)

$\phi_x \backslash \phi_y$	$-3\pi/4$	$-\pi/2$	$-\pi/4$	0	$\pi/4$	$\pi/2$	$3\pi/4$	π
$-3\pi/4$	404	156	148	140	132	128	420	212
$-\pi/2$	208	192	180	172	160	148	148	272
$-\pi/4$	240	220	208	196	188	176	164	276
0	280	248	232	220	208	196	180	180
$\pi/4$	348	300	268	256	240	224	204	200
$\pi/2$	420	384	332	308	288	264	240	228
$3\pi/4$	280	420	420	408	372	340	304	284
π	112	100	100	420	420	420	372	136

The chosen parameters in the design are dependent on the realized function.

[1] Spin Hall Effect of Double-Index Cylindrical Vector Beams in a Tight Focus, Micromachines, 2023.

[2] Actively manipulating asymmetric photonic spin Hall effect with graphene, Carbon, 2020.

[3] Spin Hall effect of transversely spinning light, Science Advances, 2022

[4] Anomalous refraction into free space with all-dielectric binary metagratings, Physical Review Research, 2020.

[5] Anomalous refraction and reflection characteristics of bend V-shaped antenna metasurfaces, Scientific Reports, 2019.

Reviewer #2 (Remarks to the Author):

The authors present a case study on vectorial-mode parity Hall effect by use of a form-birefringence metasurface resonating at optical frequencies. Experimental manipulation of the parities of two cylindrical vector beams via the designed metasurface has reassembled the well-known spin Hall effect. In addition, the authors have demonstrated parity-demultiplexed cylindrical vector beam-encoded holography. The findings are interesting and significant, and the presentation is well organized. The manuscript can be recommended for publication in Nature Communications, provided the authors consider the following points.

Reply: We sincerely appreciate the reviewer for the recognition of our work. In the following section, we are committed to providing a comprehensive and detailed response to your concerns.

1. The vectorial-mode parity Hall effect demonstrated here can be extended to the optical fiber platform beyond free space applications. Can this effect be extended to other frequency ranges, such as terahertz and microwave regimes? Are there any potential complications?

Reply: Thank you for the constructive comments. Theoretically, the mechanism behind this effect is universal and can easily be extended to any other possible frequency ranges. However, the corresponding metaatom design should be optimized for the specific frequency range to achieve the desired modulation. In response to the thoughts, we have included a short paragraph in the revised manuscript as below.

Changes made in the revised manuscript:

Page-9

...the parity of VOFs. Furthermore, the proposed parity Hall effect in this work is universal and can be extended to other frequency ranges, such as the terahertz regime and radio frequency. Therefore, the corresponding design should be adaptable to the desired frequency for achieving the desired modulation. This fine-grained manipulation of light...

2. The contrast of the measured holographic images shown in Fig. 5d appears low compared to that in Fig. 5a. It would be good if the authors can add a brief discussion on this.

Reply: We apologize for the confusion in Fig. 5d. The low contrast in the measured holographic images in Fig. 5d could be due to two reasons. Firstly, the device design incorporates inverse-vortex phases of varying orders into the metasurface to reduce crosstalk between different channels. As a result, the desired patterns (letters A, B, C, D, and numbers 1, 2, 3, 4) in the corresponding channels consist of very small focused Gaussian light spots rather than CVB spots (ring patterns), as seen in supplementary materials S4. Secondly, the illuminated area is small and it results in less clear and low-contrast detected patterns visually. The actual detected holographic pattern in a close-up picture is found reasonably better:

Fig. R1. The measured holographic Letter "A" and its amplified image.

We have revised and expanded Figure 5d in the manuscript to improve visual clarity. Please refer to the new Figure 5 in the revised manuscript.

Changes made in the revised manuscript:

Figure-5

Figure 5. CVB encoded parity-demultiplex holography. (a) Schematic of the procedure for generating a desired CVB-encoded hologram. (b) Schematic of the CVB-selective holography using a metasurface encoded hologram, along with the design of its basic super unit cell. (c) Microscopy and SEM images of the fabricated sample. (d) Experimental test results of the CVB demultiplex holography with topological charges of $m = \pm 2, \pm 3$, and even/odd parity.

3. How the aspect ratio of the TiO₂ nanostructures (up to 10 here) affects the demonstrated Hall effect, parity-demultiplexing, and meta-hologram?

Reply: In our work, the parameters of TiO₂ elliptical nanopillars are selected to achieve a full 2π -phase modulation range to facilitate the desired gradient phase modulation outlined in Eq. (3). This necessitates a comparable aspect ratio for the nanopillars to enable dipole-like resonance. The choice of a high aspect ratio of 10 in our manuscript is primarily influenced by the maturity of our fabrication process. Furthermore, we have also conducted simulations with relatively lower aspect ratios, with a height of $h=800$ nm for the nanopillars, as presented below:

Fig. R3. The scanning parameter of Dx and Dy in the situation of nanopillar height $h = 800$ nm.

It is worth noting that the desired 2π -phase modulations can be achieved within the scanning parameters range for both scenarios. Therefore, a smaller aspect ratio is feasible as long as the desired 2π -phase modulations can be fulfilled.

REVIEWERS' COMMENTS

Reviewer #1 (Remarks to the Author):

The authors have revised their manuscript, according to the comments emerged during the first review cycle. The paper can be published at Nature Communications.

Reviewer #2 (Remarks to the Author):

The authors have addressed all my questions. I recommend the acceptance of this manuscript for publication in Nature Communications.

RESPONSE LETTER

REVIEWERS' COMMENTS

Reviewer #1 (Remarks to the Author):

The authors have revised their manuscript, according to the comments emerged during the first review cycle. The paper can be published at Nature Communications.

Reply: Thank you for your encouraging feedback. We would like to express our sincere appreciation for your recommendations.

Reviewer #2 (Remarks to the Author):

The authors have addressed all my questions. I recommend the acceptance of this manuscript for publication in Nature Communications.

Reply: Thank you for your inspiring comments. We would like to express our genuine appreciation for your recommendations.